# Mouse color and wavelength-specific luminance contrast sensitivity are non-uniform across visual space

Daniel J Denman*, Jennifer A Luviano, Douglas R Ollerenshaw, Sissy Cross, Derric Williams, Michael A Buice, Shawn R Olsen, R Clay Reid

Allen Institute for Brain Science, Seattle, United States

**Abstract** Mammalian visual behaviors, as well as responses in the neural systems underlying these behaviors, are driven by luminance and color contrast. With constantly improving tools for measuring activity in cell-type-specific populations in the mouse during visual behavior, it is important to define the extent of luminance and color information that is behaviorally accessible to the mouse. A non-uniform distribution of cone opsins in the mouse retina potentially complicates both luminance and color sensitivity; opposing gradients of short (UV-shifted) and middle (blue/green) cone opsins suggest that color discrimination and wavelength-specific luminance contrast sensitivity may differ with retinotopic location. Here we ask how well mice can discriminate color and wavelength-specific luminance changes across visuotopic space. We found that mice were able to discriminate color and were able to do so more broadly across visuotopic space than expected from the cone-opsin distribution. We also found wavelength-band-specific differences in luminance sensitivity.

DOI: https://doi.org/10.7554/eLife.31209.001

## Introduction

The mouse visual system is increasingly (*Baker, 2013*; *Priebe and McGee, 2014*) being used as a model system for studying both cortical sensory processing (*Glickfeld et al., 2013*, *2014*; *Niell and Stryker, 2008*, *2010*; *Wang et al., 2011a*) and behavior (*Busse et al., 2011*; *Harvey et al., 2012*; *Histed et al., 2012*; *Hoy et al., 2016*; *Montijn et al., 2015*). While most physiological work has used achromatic stimuli (*Durand et al., 2016*; *Niell and Stryker, 2008*), mice, like most other mammals, display physiological color-opponent signals in the retina (*Baden et al., 2013*; *Baden et al., 2016*; *Breuninger et al., 2011*; *Chang et al., 2013*; *Joesch and Meister, 2016*), through LGN (*Denman et al., 2017*) and possibly V1 (*Tan et al., 2015*). The mouse retina displays asymmetric and mixed expression of its two opsins along the dorsal-ventral axis of the retina, creating opposing gradients of short and middle opsins (*Applebury et al., 2000*; *Wang et al., 2011b*) and resulting in gradients of wavelength-band-specific responses (*Chang et al., 2013*; *Denman et al., 2017*; *Rhim et al., 2017*). Therefore, the substrate for cone-driven color-opponent signals, and any color sensitivity, exists only in the overlapping 'opsin transition zone' (*Baden et al., 2013*; *Denman et al., 2017*). However, short and middle opsin responses broadly overlap in V1 and higher visual areas (*Rhim et al., 2017*) and rod-cone antagonism can also create color opponency in some mouse retinal ganglion cells (*Joesch and Meister, 2016*), presenting the possibility that behaviorally relevant color information could be extracted more broadly across retinotopic space.

Whether mice can use color information to guide visual behavior is an open question. There is some evidence for color discrimination (*Jacobs et al., 2004*), but it remains unclear how this depends on overall luminance, luminance contrast, or retinotopic position. Further, it is not known if the gradients in opsin distribution lead to variations in behavioral luminance sensitivity across space.

*For correspondence:
danield@alleninstitute.org

Competing interests: The authors declare that no competing interests exist.

**eLife digest** Color is a key part of our visual experience. Humans can distinguish between colors thanks to light-sensitive cells at the back of the eye called cones. Our eyes contain three types of cones, most simply named red, green and blue. When light enters the eye, it activates each cone type to a different degree. The combined activity of the three types of cone determines which color we see. However, in about 8% of men, one of the cone types is missing or faulty. This leads to color blindness, usually in the form of an inability to distinguish between reds and greens.

Most other mammals can also see colors. This includes mice, which are used increasingly to study the mechanisms underlying vision. But it was not clear if mice also use color to guide their behavior. Mice have only two types of cones, some of which respond to green light and others to ultraviolet light. To complicate matters, the two types of cones are distributed unevenly across the back of the mouse eye. This suggests that mice may see colors differently in different parts of a visual scene.

Denman et al. trained mice to lick a spout whenever they noticed a change in the color or brightness of a dot appearing at various locations on a screen. The results revealed that mice could detect changes in both color and brightness. But the mice's ability to do so depends on where the change occurs. In the upper part of the visual field, corresponding to the area above the horizon, mice could distinguish between different colors. In the lower part of the visual field, below the horizon, they could not. By contrast, mice were able to detect changes in brightness at many different locations.

This information will make it easier to design and interpret experiments that use mice to try to understand how the brain generates vision. This should help scientists develop new treatments for disorders that affect vision, and possibly for many forms of cognitive impairment too.
DOI: https://doi.org/10.7554/eLife.31209.002

Such non-uniformity would impact studies of visuotopically extended V1 populations, such as studies of population sparsity (*Froudarakis et al., 2014*), population correlations (*Montijn et al., 2016a*) and other notions of population coding (*Montijn et al., 2015*; *Okun et al., 2015*).

Here, we use a simple behavior, change detection, to determine where in visual space mice can discriminate changes in chromaticity and luminance at ethologically-relevant mesopic ($10^{-3}$ - 3 cd/m$^2$, [*Wyszecki and Stiles, 1982*]) luminance levels. By measuring detectability of luminance and color changes separately across elevation (spanning ~75°), we are able to generate an estimate of wavelength-specific contrast sensitivity across visual space. Mice were able to discriminate color, but only at elevations above the horizon. We find both wavelength-specific luminance and color contrast sensitivity to be dependent on retinotopic location, but that these differences in sensitivity were less dramatic than expected from the cone opsin distribution, suggesting behavioral access to differential activation of rods and cones.

## Results

### Behavioral task

To examine the psychophysical and physiological basis of mouse color vision, we first trained mice in a go/no-go change detection task (*Histed et al., 2012*) in an immersive visual stimulation environment customized for delivering stimuli in the spectral bands of the mouse short and middle wavelength opsins (*Figure 1A*; Materials and methods). We use the system here to deliver a video stimulus driven by a green and ultraviolet LED projector; for each point on the stimulus, the green and ultraviolet intensity could be independently modulated. Total luminance was in the mesopic range (<3 cd/m$^2$), over which mice are both behaviorally active (*Daan et al., 2011*) and color opponent signals have been demonstrated in the retina (*Baden et al., 2013*; *Joesch and Meister, 2016*). Briefly, under this paradigm, mice indicate that they have perceived a change in the stimulus by licking a reward spout within 1 s of the change (*Figure 1B*); subsequent licks allow reward consumption (*Figure 1C*).

Following pre-training on a luminance change detection task (see Materials and methods), we switched to change detection sessions in which the ultraviolet and green intensity, centered on the

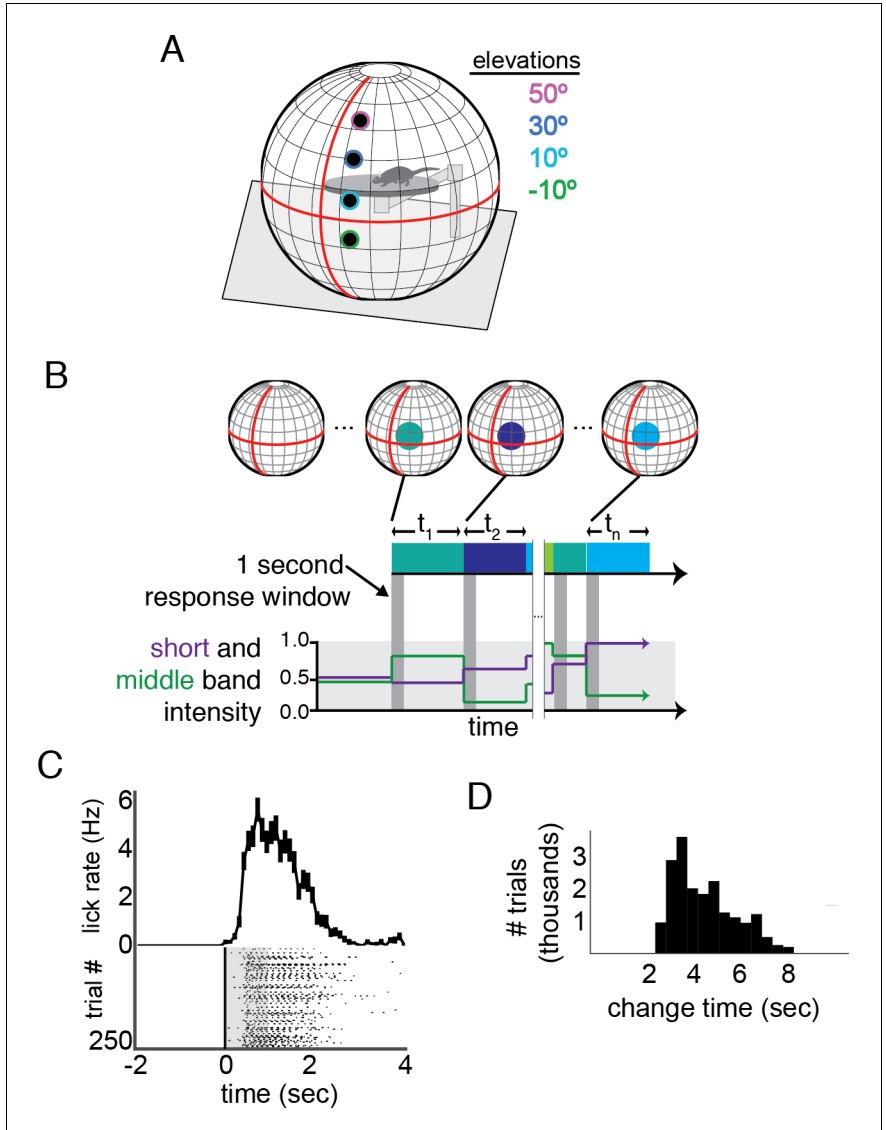

**Figure 1.** Change detection task in an immersive visual stimulation environment capable of delivering short- and middle-wavelength band stimulation. (**A**) The visual stimulation environment, with the positions and size of stimuli shown; the colored edges were not a part of the presented stimulus, but indicate the color scheme used to denote elevation throughout the other figures. (**B**) A schematic of the task. The background was set to mean intensity for each wavelength band. At variable times, $t_n$, the intensity of short and middle wavelength bands within a 15° diameter circle changed. If the mouse licked within 1 s of this change (indicate by the dark grey boxes), a reward was delivered. Schematic short and middle band intensities are shown on the lower plot; corresponding stimulus changes are shown both in blocks and cirlces above each epoch. The schematized circles are larger than actual stimuli, for clarity. (**C**) Example performance in a single session across 250 trials. Each lick is shown relative to stimulus change; the response window is overlaid in grey. A histogram of lick times is shown above. Error bars are S.E.M. (**D**) The distribution of change times (t in panel **B**) across the trials used for analysis of change detection performance. The distribution follows a log sampling distribution, enforcing roughly equal probability of a change occurring as the mouse continues to wait.

DOI: https://doi.org/10.7554/eLife.31209.003

The following figure supplement is available for figure 1:

**Figure supplement 1.** Eye position during performance of the change detection task at four elevations.

DOI: https://doi.org/10.7554/eLife.31209.004

mouse short and middle wavelength bands, respectively, were varied independently on each trial. Each trial contained a change in intensity within a 15° test circle on a mean luminance background at one of four elevations: −10°, 10°, 30°, and in some cases 50° (relative to both the horizon and the placement of the rotating mouse platform). We varied position only along elevation because both rods and cones are relatively uniform across the azimuthal axis of the retina (*Sterratt et al., 2013*). Eye position did not change with stimulus location (*Figure 1—figure supplement 1*). Stimuli at change times were modulations in ultraviolet and green intensities within the stimulus spot, resulting in changes in short and middle band stimulation. Changes in ultraviolet and green intensity were independent of each other, and of the previous intensities, resulting in mainly trials that contained luminance changes, but also some in which short and middle opsin activation oppose each to create an effective hue exchange without a luminance change. Changes covered all of the color space available using our display. There is no explicit level of chance performance in this task design, as false-positive rate could vary depending on the strategy used to guess; we include catch trials in which the stimulus did not change in order to estimate false positive rate at each elevation. To achieve sufficient trials to cover this space, we presented a total of 127,659 trials (n = 4/5 total mice trained, 284 sessions). To control for motivation, we calculated a running average of the reward rate and selected trials where this reward rate remained above four rewards per minute; only these engaged trials (44%; 56,112/127,659) were used for analysis.

## Short and middle wavelength band specific contrast sensitivity

We first examined our results to estimate the relative luminance contrast sensitivities to short and middle-wavelength band stimulation across the visual field. Although the green and ultraviolet projector LEDs nearly isolate responses of the middle and short wavelength-sensitive opsins (*Estévez and Spekreijse, 1982*), they do not necessarily isolate responses of individual cones, most of which express a combination of the two opsins. Nor do they necessarily measure the relative weight of the cone opsins themselves, as rods also may contribute to light sensitivity at these luminance levels. Rather, we present a measure of the relative perceptual weight to stimuli of the middle and short wavelength bands covered by our stimulus LEDs (*Figure 2—figure supplement 1*), as combined through both cone opsins and rods.

Total luminance change detection saturated by ~30% at all elevations (*Figure 2A*) and the half-saturation threshold (hereafter referred to as 'threshold') was less than 12% for each elevation (*Figure 2B*). The highest sensitivity was in the upper visual field, at 5.5% threshold, a threshold that is consistent with previous reports for a 15° (~0.07 cyc/°) stimulus (*Histed et al., 2012*; *Prusky et al., 2000*; *Sinex et al., 1979*; *Umino et al., 2008*). Sensitivity to increments in contrast was similar for all elevations (10.5–12.1%). Consistent with previous physiological measurements in V1 of mouse (*Tan et al., 2015*) and other species (*Kremkow et al., 2016*; *Wang et al., 2015*), sensitivity to decrements in contrast was higher than sensitivity to increments (5.4%–10.4%). Notably, this was most pronounced in the upper visual field (difference: 5%) than the lower visual field (difference: 1.3%).

To determine the independent contributions of short and middle wavelength bands, we examined change trials that contained increments or decrements of only one of the two LEDs. For the short wavelength band (i.e. UV), contrast sensitivity was non-uniform, with the highest sensitivity in the upper visual field (*Figure 2C,D*; 8% threshold). As with total luminance, mice were more sensitive to decrements than increments in contrast. The non-uniformity across elevation was more pronounced for short-wavelength-specific sensitivity than total luminance, but was restricted to decrements. The middle-wavelength (i.e. green) luminance contrast sensitivity was also non-uniform, across elevation, but with the opposite relationship as the short wavelength and total luminance: higher sensitivity at the two lower elevations and the highest tested elevation (*Figure 2E–F*). Middle-wavelength sensitivity was very similar for increments and decrements, again different than total and short-wavelength luminance contrast sensitivity. In summary, we found that luminance contrast sensitivity was non-uniform, with significant opposing wavelength-band-specific non-uniformities, although less than what would be predicted from the opsin expression or photoreceptor response alone, as shown below.

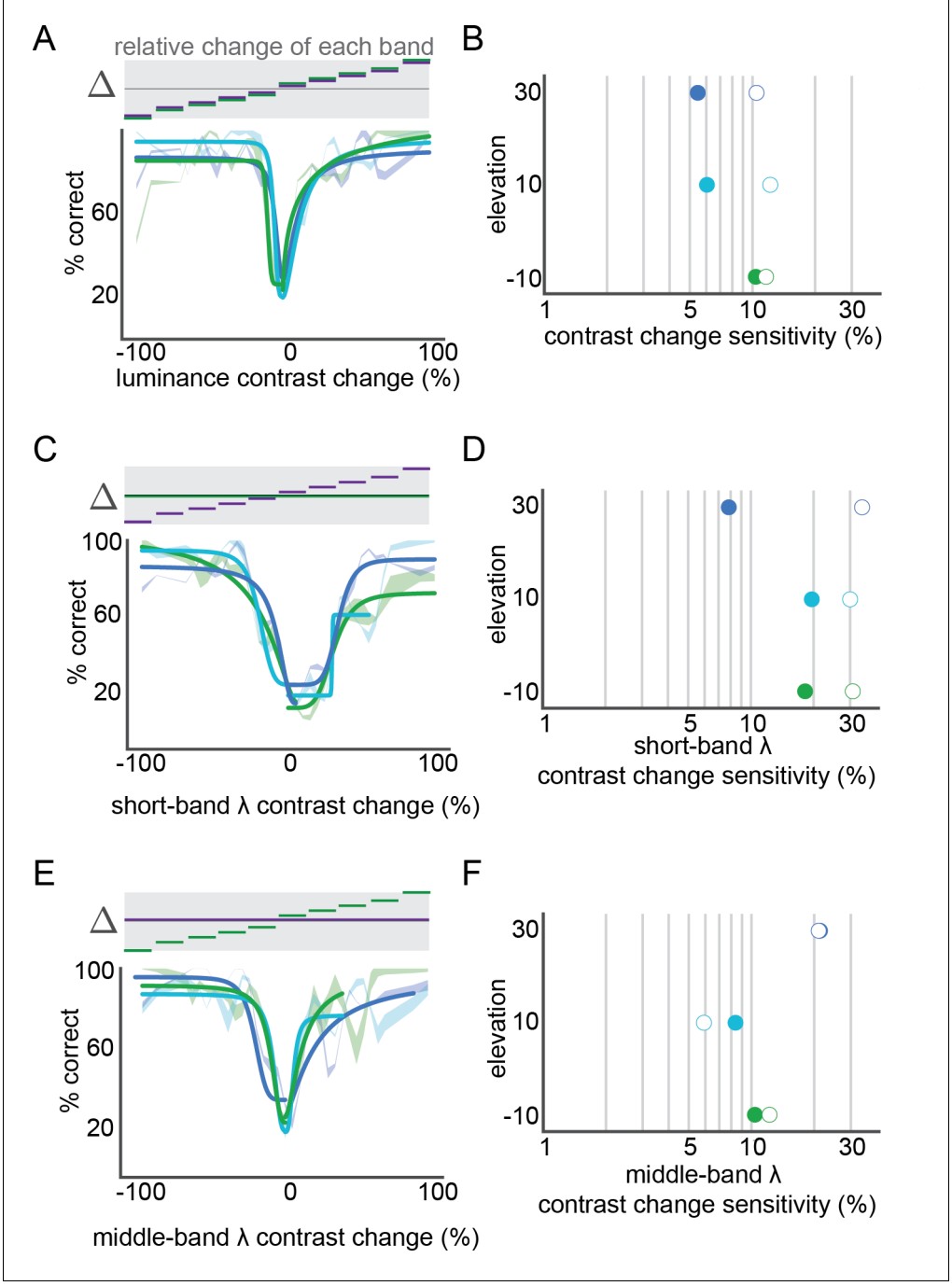

**Figure 2.** Short and middle wavelength band specific contrast sensitivity. (A) Performance of luminance contrast change detection at three elevations (green lines: −10°, light blue lines: 10°, dark blue lines: 30°). For each elevation, a fit with hyperbolic ratio function is shown overlaid on mean performance; mean performance line thickness shows S.E.M. across mice. The stimulus is schematized above the performance, showing the corresponding relative change in each wavelength band for each condition. (B) Contrast sensitivity at each elevation, from the fits in panel A; closed circles: decrements in contrast, open circles: increments in contrast. (C, D) short-band specific luminance contrast performance across elevation, as in panels (A,B). (E,F) middle-band specific luminance contrast performance across elevation, as in panels A,B.

DOI: https://doi.org/10.7554/eLife.31209.005

The following figure supplement is available for figure 2:

**Figure supplement 1.** Spectral radiance of various illumination sources from 350 to 750 nm.

*Figure 2 continued on next page*

*Figure 2 continued*

DOI: https://doi.org/10.7554/eLife.31209.006

## Determination of relative short and middle wavelength band contributions at several retinotopic locations and comparison with predicted cone weights

We next determined the relative strength of short- and middle-wavelength stimulation across the visual field. Despite existing measurements of the cone gradients and non-uniformity in V1 response (*Rhim et al., 2017*), we were uncertain about the relative contributions of rods and cones at the tested light levels (i.e. relative contributions of the rod and middle opsin to middle band stimulation), so we first determined which combinations of wavelength band activation effectively opposed each other at each elevation.

For both this determination of relative wavelength band weight, and our study of color sensitivity (see below), our approach is schematized in *Figure 3A*: for each trial, we plot the change (at the change time in the task, see *Figure 1B*) of each LED intensity against the other. Equal but opposite changes in the activation of the short and middle bands should oppose each other and lead to some change in chromaticity (colorfulness relative to luminance), but only a subset of changes will have sufficient change in chromaticity to be distinguishable (*MacAdam, 1942*), if the mouse can distinguish chromaticity at all. Unbalanced changes in intensity create a luminance change, and often a chromaticity and hue change as well, with only those that have equal changes to each wavelength band produce a pure luminance change. Because sensitivity is higher to any luminance change than pure chromaticity changes, the resulting performance forms an elongated ellipse when plotted this way. If our stimulation of each band were equal, the major axis of this ellipse would fall along the unity line, indicating equiluminant stimuli that contain chromaticity change without a luminance change. However, the relative expression of each opsin are not equal across the retina (*Applebury et al., 2000*; *Baden et al., 2013*; *Ortín-Martínez et al., 2014*), so the slope of the major axis of the observed ellipse, hereafter called the 'equiluminant line', indicates the relative weight of each luminance band at that location. As such, a slope of 1 lies on the unity and equal short and middle-band weight (*Figure 3A*, left); slopes >1 indicate middle band domination (*Figure 3A*, right) and <1 short band domination. Note that the axes of these plots are changes in band activation, not absolute intensity levels. The trials in any given bin in the plot have different absolute values of color, both before and after the change point, but the movement within any color space at the change point is always the same (in both direction and magnitude).

We measured the axis along which opposite changes in short and middle wavelength stimulation effectively canceled each other at several elevations. The performance across pairwise combinations of changes in short and middle band luminance (*Figure 3B*) was fit with an ellipse (*Figure 3C*, top), and the major axis of this ellipse was taken to be the equiluminance line (*Figure 3C*, bottom). We found the mouse to be more sensitive to middle band stimulation than expected at all elevations tested, including at 30° where, given the eye positioning (*Figure 1—figure supplement 1*), short-opsin expression dominates. In fact, surprisingly, the mice were more sensitive to middle than short-band changes at all elevations, with the following middle/short ratios: 3.4, 3.6, and 2.25 at –10°, 10°, and 30°, respectively.

We compared our measure of the wavelength-band-specific perceptual contributions to predictions from the cone expression distribution, the cone functional response, and intrinsic imaging of the mouse visual system (*Baden et al., 2013*; *Rhim et al., 2017*). By projecting (*Sterratt et al., 2013*) the spatial profile of cones into visuotopic coordinates based on estimations of the mean eye position during our experiments (*Figure 1—figure supplement 1*), we computed expected middle/short ratios for each of the elevations we tested (*Figure 3—figure supplement 1*). Similar to the estimated relative opsin weights across V1 under photopic conditions (*Aihara et al., 2017*; *Rhim et al., 2017*), 2.3, 0.81, 0.81, the cone functional distribution predicts 2.3, 1.0, and 0.44 at –10°, 10°, and 30°, respectively. Both predict far less middle-band sensitivity that we observed. This result suggests that rod opsins, centered near the middle-band, contribute significantly to mouse perceptual sensitivity at these light levels, at least as much as 60% (*Figure 3—figure supplement 1*) at higher elevations.

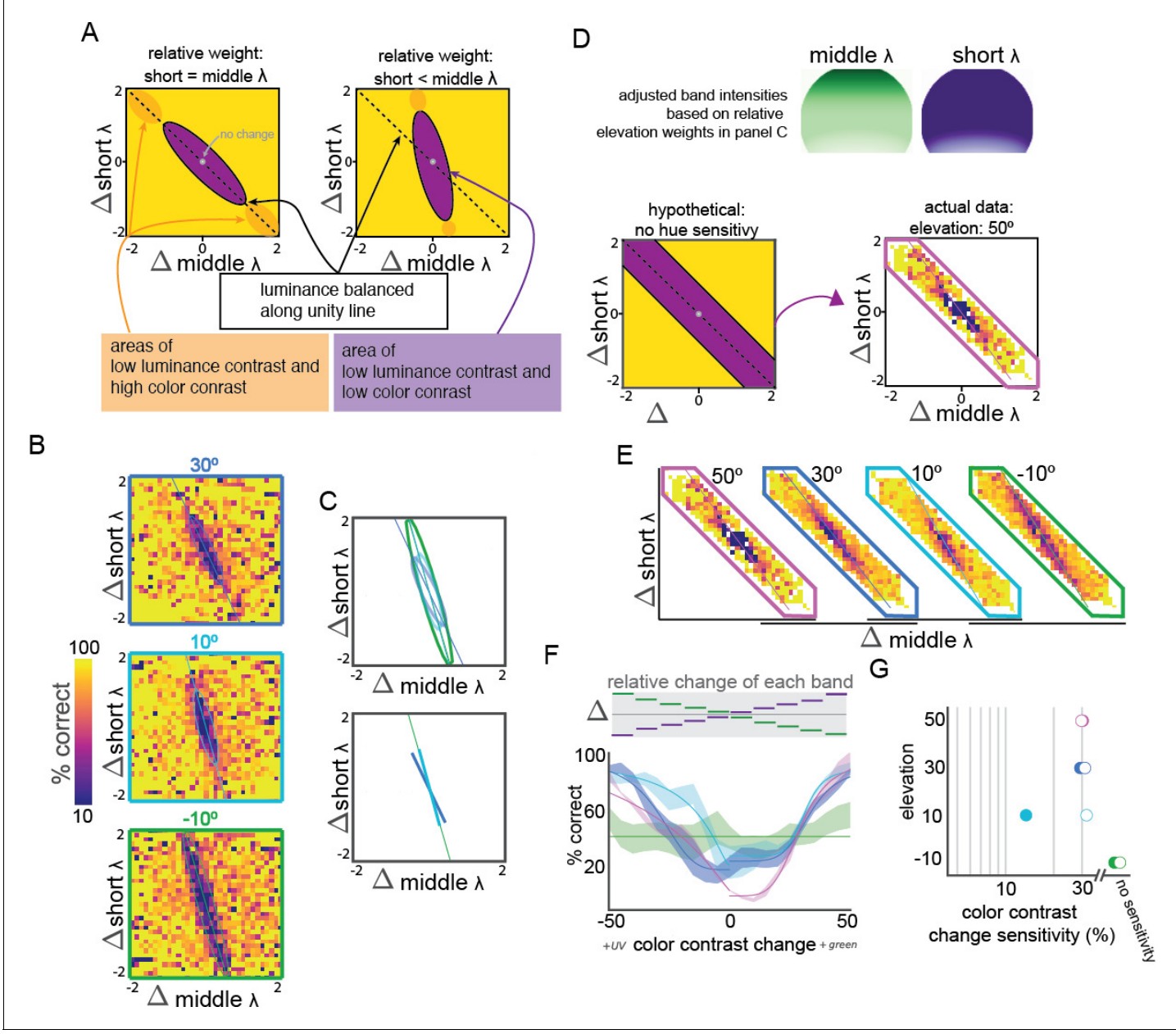

**Figure 3.** Relative short and middle-wavelength band weights and color discrimination across elevation. (**A**) Schematic representation of two possible relative short and middle band weight scenarios, including some putative color contrast sensitivity. The left scenario shows equal weight of short and middle bands, leading to balanced sensitivity and chance performance in response to equal and opposite changes in short and middle band contrast, along the unity line. The right scenario shows higher middle band sensitivity, resulting in a positive shift of the slope, where the larger changes in short band contrast are required to balance changes in middle band contrast. For both scenarios, low luminance and hue contrast should yield low change detection performance (purple ellipses). Any hue discrimination occurs along the major axis of this purple ellipse, and color contrast is high near the edges (orange ellipses). Axes indicate change in each wavelength band intensity, combining the same relative changes in color and luminance across many absolute hue, chromaticity, and luminance values. (**B**) Observed relative weights of short and middle band weights, as in panel **A.**, at elevations of −10° (green outline, bottom), 10° (light blue outline, middle), and 30° (dark blue outline, top). (**C**) Fit of the relative weights in panel **B** with two-dimensional Gaussians, including a plot of the major axis of the fit. These major axes are isolated in the plot below. Color corresponds to stimulus elevation (green lines: −10°, light blue lines: 10°, dark blue lines: 30°). (**D**) Schematic of sensitivity testing after adjusting for the wavelength band weights in panels **B** and **C** (top panels showing adjusted intensities across elevation, normalized to the non-adjusted values. Following this adjustment, the central region should be along the line of equiluminance (bottom left), so subsequent testing was focused on this spear-like region (arrow to bottom right). Actual data at elevation = 50° same as panel **E**. (**E**) Performance of change detection at four elevations (green lines: −10°, light blue lines: 10°, dark blue lines: 30°, pink lines: 50°) after short and middle weights adjustment. Because short and middle weight were not exactly balanced, performance was fit with a two-dimensional Gaussian and color sensitivity measured along the major axis (lines overlaid on each plot). (**F**) Hue sensitivity

*Figure 3 continued on next page*

*Figure 3 continued*

at each elevation. Fit with hyperbolic ratio function is shown overlaid on mean performance; mean performance line thickness shows S.E.M. across mice. The stimulus is schematized above the performance, showing the corresponding equal and opposite relative change in each wavelength band for each condition. (G) Contrast sensitivity at each elevation, from fits in panel F. Closed circles: decrements in contrast, open circles: increments in contrast.

DOI: https://doi.org/10.7554/eLife.31209.007

The following figure supplement is available for figure 3:

**Figure supplement 1.** Estimating short and middle cone weights in the coordinates of our behavioral apparatus from retinal expression and functional data.

DOI: https://doi.org/10.7554/eLife.31209.008

## Can mice discriminate color?

We continued to ask if mice could report a change in chromaticity independent of any luminance change. Instead of explicitly creating a device that normalized total luminance during chromaticity changes (*Gagin et al., 2014*; *MacAdam, 1942*; *Wyszecki and Stiles, 1982*), we presented sufficient combinations of wavelength-band-specific luminance changes to experimentally determine when chromaticity changes occurred independent of luminance changes. We began with the approach (and data) described above for *Figure 3A–C*, but now specifically examining which combinations indicate color discrimination independent of luminance change. Again note that the axes of these plots are changes in band activation, not absolute intensity levels, so the major axis of the ellipse is equivalent to chromaticity changes and the minor axis lightness change, collapsed across all hues.

We start with the assumption that a lack of change in luminance or chromaticity is not discriminable to the observer. All luminance and color contrast changes that are behaviorally indistinguishable from this 'no change' condition (the 0,0 point in *Figure 3A,B*) form an ellipse of non-discriminability analogous to a MacAdam ellipse of human non-discriminability on the human chromatcity diagram (*MacAdam, 1942*); in our case, this ellipse is in a chromaticity-luminance space (*Figure 3A, B*). As noted above, the major axis of this ellipse specifies this equiluminance line for the wavelength band sensitivities at that elevation). This line is equivalent to a slice through the equiluminant plane in a DKL color space (*Derrington et al., 1984*), and by examining change detection performance along this experimentally defined axis of chromaticity change, we can ask if mice can discriminate color independent of luminance (*Figure 3A*, left, orange ellipses).

Indeed, qualitative inspection of the extremes of the equiluminance lines in *Figure 3B* indicated color discrimination ability at some elevations, and differences across visuotopic space. However, because of the shift of short versus middle contributions across space (slopes of the lines in *Figure 3C*), we were concerned that the stimuli along the measured equiluminance line at each elevation (*Figure 3B*) could contain a within-stimulus luminance gradient that allowed the mice to exploit small within-stimulus luminance changes to detect changes we interpreted as color changes. Further, because the range of chromaticities presented at each elevation was unequal (lengths of the lines in *Figure 3C*), we were concerned that the mouse could be sensitive to chromaticity changes at −10° outside of the range of the stimuli presented. To overcome these issues, we adjusted the intensity of the middle band stimulation by fitting an exponential to the slopes of the equiluminance lines across space, normalizing this to the maximum possible intensity of the green LED (*Figure 3D*, top). This should create conditions of uniform luminance contrast across all elevations, shifting the major axis of the ellipse at all elevations to the unity line (*Figure 3D*, bottom). As such, we limited our testing conditions to the changes around the unity line in order to reduce the number of sessions needed to achieve sufficient trials in each condition; this results in spear-like plots focused on luminance-independent color change. While our adjustment failed to perfectly compensate for variable short-middle weights, we were still able to capture more chromaticities at each elevation and minimize potential luminance gradients.

We found color discrimination to depend on elevation. Examining the adjusted data (*Figure 3F–H*), we found color discrimination was negligible at −10°, but mice were capable of varying levels of color sensitivity at all other elevations tested (*Figure 3F,G*). The performance along the equiluminant line at −10° was not well fit by a hyperbolic ratio function, and the performance in catch trials (0% contrast change, false-positive rate) was not significantly different from any point along the line

(p>0.05, student's t-test). We were able to fit the performance at each of the other elevations tested, up to 50° above the horizon. Color contrast sensitivity was highest for decrements in short-middle opponency at 10° elevation (13.4%); color sensitivity was nearly identical for decrements in short-middle opponent contrast at 30 and 50° and for increments at all elevations above the horizon (29.3–32.1%). Notably, the false-positive rate (0% contrast change, catch trials) was markedly different between the lowest and other elevations. We interpret this is a shift in strategy in order to maximize reward volume, as the total volume of reward did not differ between −10° and the other elevations (0.8−0.92 mL/session, p>0.2, Welch's t-test), despite the lack of color discrimination at −10°.

## Discussion

The finding that both wavelength-specific luminance (*Figure 2*) and color contrast (*Figure 3E–F*) sensitivity is not uniform across the visual field is in accordance with the distributions of both retinal and primary visual cortical (*Rhim et al., 2017*) responses. However, we found that middle-band sensitivity was both higher and more uniform than expected (*Figure 2*). This suggests that rod sensitivity contributes significantly to perceptual sensitivity at these light levels (*Figure 3—figure supplement 1*).

This finding may be important for studies of the mouse visual system that use visuotopically extended stimulation (*Froudarakis et al., 2014*; *Garrett et al., 2014*; *Montijn et al., 2015*; *Montijn et al., 2016b*; *Pouille et al., 2009*; *Zhuang et al., 2017*), especially those that measure the underlying population representation of the stimuli. Because the spatial scale of luminance and contrast adaptation can be large (*Smirnakis et al., 1997*), the adaptation to large single-band stimuli (such as those produced by LCD or other sRGB displays) in these studies may underestimate the contrast sensitivity for cells in upper visual field. This spatial scale is especially relevant because of the scale of mouse vision – 50% differences can be seen across a small number (~5) receptive field diameters.

Our results also demonstrate that color sensitivity depends on retinotopy, and that some retinotopic locations appear to not support color discrimination. A goal of many large-scale data collection efforts, both completed (*Baden et al., 2016*) and underway (brain-map.org/visualcoding) as well as smaller-scale surveys (*Durand et al., 2016*; *Gao et al., 2010*; *Niell and Stryker, 2008*; *Piscopo et al., 2013*) from retina to V1 is the classification or clustering of response properties in order to define functional channels. Because color-opponent cells, both single and double (*Shapley and Hawken, 2011*), are thought to underlie such behavior, our findings indicate that mice may have at least one, likely at least two, color-opponent cell types; the presence of such functional cell types may depend strongly on retinotopy. Notably, our animals are housed in an environment with fluorescent lighting that does not provide UV-B for reflection (*Figure 2—figure supplement 1*), suggesting that the behavior we observed is developmentally specified, not learned, and further not lost through lack of use.

Any color-opponency depends on the underlying distribution of opsins in the retinal photoreceptor layer. The distributions of cone opsins in the mouse retina has been well described, with only a subset of pure cones expressing only a single opsin and 40% of cones expressing a mix of both, and a non-uniform distribution of both pure and mixed cones: middle-wavelength opsin dominates the dorsal retina, although it also extends ventrally, while the short-wavelength opsin is more constrained to the ventral retina. (*Applebury et al., 2000*; *Baden et al., 2013*; *Ortín-Martínez et al., 2014*; *Wang et al., 2011b*). In total, cones are relatively uniform across the dorsal-ventral axis of the retina, although others report some increased density in the ventral retina. Rods, on the other hand, are uniform across the dorsal-ventral axis of the retina and are much more numerous than cones (*Jeon et al., 1998*). If color responses, both physiological and behavioral, rely only on cone signals they should follow the cone opsin distribution exactly: more in the transition zone, some in the upper visual field, and none in the lower visual field. Our results are consistent with cone-cone opponency, as we observed a lack of color discrimination only in the lower visual field. However, our result also indicates some rod-cone opponency, as color change detection behavior thresholds relatively unchanged across 10° to 50°, despite the steep decline in middle opsin expression across this range of the retina (*Baden et al., 2013*).

Mice, while often considered nocturnal (*Febinger et al., 2014*), can be behaviorally active across a range of luminance conditions; C57BL/6 mice in particular can shift between diurnal, crepuscular, and nocturnal behavioral patterns over the course of the year (*Daan et al., 2011*). Previous studies on color signaling in the mouse have offered several hypotheses for the ethological uses of color signals. Our results are consistent with the hypothesis (*Baden et al., 2013*) that non-uniform wavelength-specific sensitivity is matched to the luminance statistics of natural scenes (UV in the upper visual field, green in the lower), and high sensitivity to decrements in the short wavelengths may be particularly helpful during the shift toward UV in the spectral radiance distribution the during twilight hours (*Spitschan et al., 2016*). Another hypothesis, that short-middle opponency is useful for identifying mouse urine posts (*Joesch and Meister, 2016*), is inconsistent with our demonstration of a lack of color discrimination at −10°, at least in absence of significant excursions in eye position or head movements. Our results suggest that, under the mesopic condition during which mice are often active, color signals may be mediated by both cone-cone and some rod-cone opponency, and this may facilitate the specialization of cone opsin distributions for sampling natural luminance statistics (*Baden et al., 2013*; *Chiao et al., 2000*).

## Materials and methods

All procedures were approved by the Allen Institute for Brain Science Institutional Animal Care and Use Committee.

### Animals and surgical preparation

All animals used in this study (n = 5) were C57Bl/6J male mice aged 30–300 days obtained from The Jackson Laboratories (IMSR_JAX:000664). To fix the animal's head within the behavioral apparatus, a single surgery to permanently attach a headpost was performed. During this surgery, the animal was deeply anesthetized with 5% isoflurane and anesthesia maintained throughout the surgery with 1.5–2% continuous inhaled isoflurane. The mouse was secured in a strereotax with ear bars; hair was removed and the exposed skin sterilized with three rounds of betadine. An anterior-posterior incision was made in the skin from anterior of the eyes to posterior of the ears. The skin was removed in a tear drop shape exposing the skull. The skull was leveled and the headpost was placed using a custom stereotaxic headpost placement jig. A custom 11 mm diameter metal headpost with mounting wings was affixed to the skull using dental cement. The exposed skull inside the headpost was covered with a thin protective layer of clear dental cement and further covered with Kwikcast. The animal was allowed to recover for at least 5 days prior to the initiation of behavioral training.

After headpost implantation, animals were kept on a reverse light cycle (lights OFF from 9AM to 9PM) and behavioral testing was done between 9AM and 1PM. Mice were habituated to handling gradually, through sessions of increasing duration. Mice were also habituated to the behavioral apparatus, first by allowing periods of free exploration and subsequently with head fixation sessions increasing from 10 min to 1 hr over the course of 1 week. Water restriction began with habituation; all mice were maintained at 85% of the original body weight for the duration of training and testing.

### Stimulus environment and stimuli

Ultraviolet and human-visible stimuli were provided across a range of retinotopic locations using a custom spherical stimulus enclosure (*Denman et al., 2017*) (*Figure 1A*, *Figure 2—figure supplement 1*). A custom DLP-projector designed for the mouse visual system provided independent spatiotemporal modulation of ultraviolet (peak 380 nm, *Figure 1B*) and green (peak 532 nm, *Figure 1B*) light. The projection system operated at 1024 × 768 pixel resolution and a refresh rate of 60 Hz, achieving a maximum intensity of ~3 cd/m2. Planar stimuli were spatially warped according to a custom fisheye warp for presentation on a curved screen; the fisheye warp was created through an iterative mapping protocol using the meshmapper utility (http://paulbourke.net/dome/meshmapper/) calibrated on the behavioral environment to achieve maximal accuracy.

Stimuli were presented in the right visual hemifield and consisted of 15° diameter circles of varying color on a mean intensity background using custom written software extensions of the PsychoPy package (http://www.psychopy.org, RRID:SCR_006571). The background intensity was 1.52 cd/m$^2$. For some testing sessions, the color of the display was adjusted to match the mouse's spectral sensitivity in order to create uniform and balanced sensitivity to the projector's LED sources across the

visual stimulus enclosure. To do so, a second custom warp was applied that included a spatially dependent adjustment of the intensity of each LED (near-UV and 'green'), according to the results shown in *Figure 3C*.

Animals were head-fixed on a freely rotating disc in the center of the spherical enclosure and allowed to run freely during the course of training and testing. A lick spout was positioned approximately 0.5 cm in front of the mouse within range of tongue extension.

In some experiments, infrared short-pass dichroic mirrors (750 nm short-pass filter, Edmund Optics) were placed in front of each eye to allow for video tracking of the pupil. Cameras (Mako and Manta, AVT technologies) placed behind the animal were aligned to record a reflected image of the pupil; infrared illumination and a reference corneal reflection was provided via an LED positioned near the camera. Movies of the eye position during presentation of the stimuli used in the task was acquired at >=60 Hz, with the eye occupying >60% of the image at 300 × 300 pixels. Data from these sessions were not included in the performance analysis to avoid any potential artifact caused by the infrared dichroic.

## Behavioral task

Animals were first shaped to associate changes in luminance with a reward. After each change in luminance, a water reward was automatically delivered, regardless of mouse licking behavior. During these sessions, the reward was constant at 10 μL. Incorrect licks were punished by resetting the trial, such that the mouse had to wait longer for the next change. This 'shaping' phase lasted a minimum of 2 days, but for most mice extended to several weeks. For some animals (2/5), subsequent epochs of this automatic reward shaping served as task reinforcement when performance in testing blocks dropped.

During each testing session, a circle was presented at a single visuotopic location and remained at this location for the duration of the session. At non-regular intervals, again selected from an exponential distribution, the color and/or luminance of the stimulus was changed, and the mouse had to report detection of change by licking the reward spout within 1 s of the change in order to receive reward (*Figure 1C*). Licks were detected through a capacitive sensor connected to the reward spout. No water was present on the reward spout before the first lick; if the animal correctly detected a change, a water reward (3–10 μL, depending on animal and stage of training) was delivered through this spout (*Figure 1D*). Sessions were 50–60 min and typically included ~300 trials. The percent correct reported throughout this work is the total number of trials of that condition in which the mouse licked within 1 s of the change, divided by the total number of trials of that condition. The percent correct on catch trials, in which no stimulus change occurred but rewards were delivered for licks within 1 s of fictive change, is interpreted as the false positive or 'chance' performance level.

Mice were first trained to associate changes of a 15° stimulus at 100% luminance contrast with a reward. In these sessions (total of 3 to 25 sessions), the contrast of a stimulus (10° elevation, relative to the horizon) changed at exponentially distributed intervals from 50% positive relative to the background to 50% negative (from white to black), or vice versa (black to white). If a lick occurred within 1 s of an actual stimulus change (*Figure 1B*), a reward was delivered to the spout and liquid reward was consumed subsequent licks (*Figure 1C*). If a lick occurred outside of this window the trial was aborted, extending the time the mouse must wait and effectively creating a 'time-out' period. Mice advanced from this protocol after performance exceeded 75% for consecutive sessions.

In subsequent testing sessions, the intensity of the ultraviolet and green intensities were varied independently on each trial. Each trial contained a change in LED intensity for a 15° test circle on a mean luminance background at one of four elevations: −10°, 10°, 30°, and in some cases 50°. The first 8–20 trials of each session were 100% contrast changes, as described for the training blocks, with rewards automatically delivered. The number of these daily 'free' rewards was reduced to eight for as long as the mouse received >1.0 mL of reward during training or performed well enough to reach satiety and disengage from the task. We attempted to correct for sessions with poor performance by increasing these 'free' rewards on subsequent days before gradually reducing them again. To control for motivation in the results, we calculated a running average of the reward rate and selected trials where this reward rate remained above four rewards per minute; only these engaged trials (44%; 56,112/127,659) were used for analysis.

## Analysis

All analyses were done using Python (RRID:SCR_008394) and common scientific packages (numpy RRID:SCR_008633, scipy RRID:SCR_008058, matplotllib RRID:SCR_008624, and pandas). Code is publicly available (*Denman, 2017*) and includes a Jupyter notebook that contains code for generation of our figures from the data. Data from each training session was saved and combined into a common data structure, also available from *Denman, 2017*, that was used for all analysis. Individual sessions were analyzed to drive adjustments in the training parameters such as the number of automatic 'free' rewards. Following data collection for all animals, all sessions were loaded into a single object for analysis. This data structure can be recreated from the files made available from https://github.com/danieljdenman/mouse_chromatic (copy archived at https://github.com/elifesciences-publications/mouse_chromatic).

To quantify performance, from each trial the following parameters were extracted: change times (the time of stimulus change), lick times (the time of each lick, as detected through the capacitive sensor connected to the reward spout), and the stimulus conditions. A trial was scored 'correct' if the first lick after a change time occurred with one second, and if there was actually change in intensity of either green or ultraviolet at that change time.

For each mouse, the percent correct was computed for each pair of LED state transitions, that is, each pairwise combination of change in short-band luminance and change in middle-band luminance (e.g. *Figure 3C*). For each mouse, performance was ignored if three trials were not presented for those conditions. For fitting, missing data were replaced via a nearest neighbors approach, with the mean of the surrounding data. Our sampling strategies focused on the areas of changing performance, ensuring that cases of missing data were limited to the areas where performance had saturated at or near the lapse rate. Psychophysical curves for wavelength band-specific and color sensitivity were taken from the appropriate slices of this color space. Sensitivity was taken from the $c_{50}$ parameter of fit a hyperbolic ratio fit (*Contreras and Palmer, 2003*) with an additional offset term, $R_0$, to account for shift in false-positive rate across conditions.

A total of five mice entered training on the task; one mouse failed to reach consecutive sessions of 75% performance during the initial high luminance contrast change detection phase, and so did not continue to testing in the color contrast discrimination phase. We did not use any statistical methods to determine mouse or trial sample size prior to the study, determining based on stability and consistency of results when sufficient samples had been collected. Statistical tests were student's t-test unless otherwise specified.

Eyetracking analysis was done via a semi-automated starburst algorithm; full details are available from the Allen Brain Observatory Visual Coding Overview v.4, June 2017 <http://help.brain-map.org/display/observatory/Documentation>. Briefly, the algorithm fits an ellipse to the pupil or corneal reflection (CR) area within user specified windows outlining the location of the pupil and corneal reflection.. A seed point is identified by convolution with a black square (for the pupil) and white square (for the corneal reflection). An ellipse was fit to candidate boundary points identified using ray tracing using a random sample consensus algorithm. The fit parameters were first reported in coordinate centered on the mouse eye. To convert the tracked pupil positions from pixel coordinates in the eye tracking images to visuotopic coordinates, we first converted the pixel coordinates to spherical coordinates using an orthographic projection of the image onto a sphere with a radius of 1.16 mm, the reported radius if the mouse eye. Because the mouse was positioned in the center of the stimulus enclosure (see the diagrams in *Figure 1—figure supplement 1*, panel A), which is also a sphere, the position of the pupil in these 'eye' spherical coordinates can be converted to visuotopic coordinates by projecting the center of the 'eye' sphere on to the visuotopic sphere. The reflection of an infrared LED off of the cornea indicated the center of each eye, and we use the placement of each mirror relative to the mouse (right eye: 51.2° in azimuth, 0° in elevation; left eye: 60° in azimuth and −10° in azimuth) to determine the direction of gaze. As such, the corneal reflection in each eye tracking movie indicates a known reference point in visuotopic space, determined by the relationship of the imaging plan to the eye sphere, and the difference between the corneal reflection and the pupil allows for estimation of gaze position in visual coordinates. Coordinates for eye position were extracted independently for each frame of the eye position movie. This calculation assumes that the center of both eyes are at the center of the visual stimulation dome, that the each eye is a sphere, and that the movements of the eye are rotations about the center, none of which is

strictly true. While left and right eye movies had different contrast and noise levels (see *Figure 1—figure supplement 1*, panels C-D), estimations of eye position in visual degrees brought disparate pixel measurement from each eye into good agreement with each other (compare y pixel measures to elevation in *Figure 1—figure supplement 1*, panel C). We believe this calculation to be accurate enough for comparison of relative eye positions and, given the small displacement of the eyes from the center (<1 cm) of the dome (30 cm radius), reasonable for using published retinotopic distributions to estimate relative opsin distributions in visuotopic coordinates.

## Acknowledgements

We would like to thank Saskia de Vries, Justin Kiggins, and Brian Long for useful discussions in the preparation of this manuscript. We would also like to thank the Neurosurgery and Behavior team, Animal Care team, and Naveen Oullette for assistance in animal surgery, care, and handling. We wish to thank the Allen Institute founders, Paul G Allen and Jody Allen, for their vision, encouragement and support.

## Additional information

### Funding

| Funder | Author |
| --- | --- |
| Allen Institute for Brain Science | Jennifer A Luviano<br>Douglas R Ollerenshaw<br>Sissy Cross<br>Derric Williams<br>Michael A Buice<br>Shawn R Olsen<br>R Clay Reid<br>Daniel J Denman |

The funders had no role in study design, data collection and interpretation, or the decision to submit the work for publication.

### Author contributions

Daniel J Denman, Conceptualization, Data curation, Software, Formal analysis, Supervision, Investigation, Visualization, Methodology, Writing—original draft, Writing—review and editing; Jennifer A Luviano, Sissy Cross, Investigation, Writing—review and editing; Douglas R Ollerenshaw, Michael A Buice, Software, Methodology, Writing—review and editing; Derric Williams, Software, Writing—review and editing; Shawn R Olsen, Software, Supervision, Methodology, Writing—review and editing; R Clay Reid, Conceptualization, Supervision, Writing—original draft, Writing—review and editing

### Author ORCIDs

Daniel J Denman ⓘ http://orcid.org/0000-0003-1075-1265
R Clay Reid ⓘ https://orcid.org/0000-0002-8697-6797

### Ethics

Animal experimentation: All procedures were approved by the Allen Institute for Brain Science Institutional Animal Care and Use Committee (IACUC, protocol 1506).

### Decision letter and Author response

Decision letter https://doi.org/10.7554/eLife.31209.011
Author response https://doi.org/10.7554/eLife.31209.012

## Additional files

### Supplementary files
• Transparent reporting form
DOI: https://doi.org/10.7554/eLife.31209.009

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
