## [Decision Letter]

Thank you for submitting your article "Mouse hue and wavelength-specific luminance contrast sensitivity are non-uniform across visual space" for consideration by *eLife*. Your article has been reviewed by three peer reviewers, and the evaluation has been overseen by a Reviewing Editor and David Van Essen as the Senior Editor. The following individuals involved in review of your submission have agreed to reveal their identity: Ed Callaway (Reviewer #1); Bevil R Conway (Reviewer #2).

The reviewers have discussed the reviews with one another and the Reviewing Editor has drafted this decision to help you prepare a revised submission.

Summary:

This paper addresses the extent to which mice have color vision. Although it is known that mice have two different cone types, it is not completely clear that mice use the retinal signals to extract color information. The cone types are not intermingled across the retina as in trichromatic primates, but rather arranged in a counter-gradient across the dorsal ventral axis; moreover, many cone photoreceptors express more than one cone photo-pigment, again unlike trichromatic primates. These observations, documented previously, suggest that signals from the different cone types are not contributing to color vision operations analogous to those found in trichromatic mammals (or other dichromatic mammals), but used instead to support specialized behaviors such as tracking urine, or spatial navigation. Nonetheless, pioneering work by Jacobs and colleagues demonstrated luminance-invariant color discrimination behavior in mice, supporting the idea that mice are capable of some dichromatic color vision. The main advance of the current report is a thorough quantification of the color abilities of mice across the topography of the retina. The experiments are very ingenious and well designed, especially given the challenge of obtaining psychophysical measurements from head-fixed mice, and the requirement of stimulating precise elevations over the visual field. Thus, the paper presents major technical innovations that couple eye tracking and careful color-psychophysical tests in alert mice. The work is heroic, requiring many hours of animal-behavior testing. The results will be of interest to the growing number of laboratories that are investigating neural and molecular mechanisms supporting vision in mice. The results are surprising: the color abilities assessed in the paper do not correspond to predictions based on the gradients of cone pigments. The authors speculate that this discrepancy reflects a substantial contribution of rods to the behavioral tests, which is supported by recent physiological evidence of rod-cone opponency in mice from Meister's group.

Essential revisions:

Despite the expertise of the reviewers, all three had some trouble following the descriptions of the visual stimuli used to assess hue sensitivity and the related presentation of results in Figure 3. Because each reviewer had somewhat different comments, rather than synthesizing them, they are all copied below to show the different sources of confusion. You will therefore find some redundancy in these comments.

Another essential revision will be to address concerns about measurements of eye position as summarized below.

1) Visual stimuli, hue sensitivity and Figure 3:

Reviewer 2 –

The stimulus is not completely clear to me. On a conventional CRT one might have a neutral adapting background and replace a section of the background with a stimulus spot through silent substitution; alternatively, one might superimpose the stimulus on the background, so that the stimulus has the mixture of background and stimulus (which might be used to selectively adapt a cone type). It's not entirely clear from the description of the stimulus which method is being deployed. Also, it would be conventional to determine for the stimuli the expected photoreceptor activation (the percentage photoreceptor contrast). The authors estimate the pigment density at the retinal locations of their stimuli, so presumably it should be possible to also report the cone (and rod) contrast for the stimuli in the various figures.

Figure 3 contains the main result, but it is a little confusing. I realize Figure 3 is a schematic, but it would help if there were some units, or at minimum, the location of (0,0). Is it the case that the (0,0) is at the center of the purple ellipse, and not at the bottom left corner (as would be typical for Cartesian coordinates)? The text states that the major axis of the ellipse corresponds to a line of "constant saturation and lightness but varying hue" in HSV space. I don't think this is accurate. If the purple ellipse passes through (0,0), then it would pass through a point of zero saturation (at least if I am correctly interpreting the top panel of Figure 3).

Figure 3. It is not clear how "attempted compensation for relative short and middle band weights" was performed, and specifically how this is reflected in the graph (how is Figure 3 different from Figure 3?).

In Figure 3, is 50% correct equivalent to "chance"? I'm inclined to believe this is so, but then what is the interpretation for the significant performance below chance at several elevations (and for equiluminant stimuli of low color contrast)? And in the case where my interpretation is wrong, then how does one square the significant performance at the lowest elevation, at all color-contrast changes, with the performance on catch trials?

Figure 3, shows that there is a set of stimuli defined by a ratio of S increment/M decrement (or M increment/S decrement) that is relatively invisible to the animals. The balance point of S/M is slightly different across the retina, with the more ventral visual field (dorsal retina) showing greater sensitivity to M than S, compared to the dorsal visual field. This suggests that the dorsal retina has a higher M:S ratio, which I believe is consistent with the cone-opsin distribution. Is this interpretation of the results correct? If so, aren't the behavioral data more consistent with the opsin distribution than the conclusions drawn by the authors?

Reviewer 3 –

I found the last section of the Results, with the description of Figure 3 and how hue sensitivity was measured, quite unclear in several parts (see my specific "minor" comments below). This description should be more detailed and much clearer, in order for a reader to fully grasp the meaning and implications of their findings.

I have trouble understanding the experiments and data analysis to measure the sensitivity to hue changes Figure 3). Here is what I think I have understood and what I have not:

The authors want to test if mice can detect changes along the diagonal line of perceived equiluminance (i.e., simultaneous and opposite changes of luminance in the two channels that are not detected by the rats as an overall change of luminance). Since, as they show in Figure 3, the perceived equiluminant line has not slope = 1, then the authors use this perceived line to measure hue sensitivity along it.… is this correct? Is this what they meant with Figure 3? Or, instead, did they actually show combinations of the two channels that are along the unity line? In any event, I don't understand why the shape of the purple bar changes when going from Figure 3 to Figure 3. Shouldn't the purple region have the same spear-like (not rectangular) shape of the region in Figure 3 (and of the data shown below in Figure 3)? More importantly, about the color matrixes shown in Figure 3: shouldn't these data just be the same of those already shown in Figure 3 (just taking those points along the equiluminant line)? What is the difference? Are these actually data resulting from different experiments? And if so, why? Finally, are the curves shown in Figure 3 derived from the data matrixes of Figure 3? Are the curves an average of the data around the unity line? Or around the perceptual equiluminant line? If so, how many data points of the matrixes in Figure 3 are used to obtain the curves of Figure 3. In summary, this whole analysis needs far more details and explanations to be fully understandable?

2) Comments about measurement of eye position:

The data suggest that the paradigm provides an accurate assessment of mouse visual ability. It is conceivable that the false-positive (false-alarm) response rate is not fixed, but changes depending on the detectability of the stimulus. It's not clear how this (or shifts in criterion) would impact the results. The authors show that the mice do not have systematically different eye positions across trial types, but it is important to show that the experimental technique could recover differences in eye positions if these differences existed. What is the resolution of the eye tracking compared to the measured eye movements made by the animals in the experiment?

The authors mention that they have used an eye tracker, which, I believe, is essential to allow them to make accurate predictions of perceptual sensitivity, based on the known opsins' retinal distributions. However, as I far as I understand, the eye tracker was not used systematically, but only in some control sessions, whose data were not those used to obtain the sensitivity measures. I wonder then how the authors can be sure about the position of the mouse eye during the main experimental sessions (it may change form session to session or during a session, making their prediction unreliable). In addition, the authors do not provide any details (or any specific reference) about the eye tracking method they use, especially how they calibrated the eye tracker. Such a calibration is always challenging in rodents (since they do not saccade to target locations over the stimulus display). The authors report that they are able to obtain measurers in degrees, but they do not explain at all how they convert pixels to degree (these clarifications about the eye data are essential if we have to believe their predictions).

---

## [Author Response]

Essential revisions:Despite the expertise of the reviewers, all three had some trouble following the descriptions of the visual stimuli used to assess hue sensitivity and the related presentation of results in Figure 3. Because each reviewer had somewhat different comments, rather than synthesizing them, they are all copied below to show the different sources of confusion. You will therefore find some redundancy in these comments.Another essential revision will be to address concerns about measurements of eye position as summarized below.1) Visual stimuli, hue sensitivity and Figure 3:Reviewer 2 –The stimulus is not completely clear to me. On a conventional CRT one might have a neutral adapting background and replace a section of the background with a stimulus spot through silent substitution; alternatively, one might superimpose the stimulus on the background, so that the stimulus has the mixture of background and stimulus (which might be used to selectively adapt a cone type). It's not entirely clear from the description of the stimulus which method is being deployed.

We apologize that the stimulus was not clear, and have added sentences describing the stimulus in better detail to beginning of the Results section. Neither method is exactly being employed here. The stimulus replaces a constant background, as you describe, but the change in each “primary” (green and UV LEDs in the projector) is random at each replacement. We do not explicitly isolate specific photoreceptor types [especially as many mouse cones express multiple opsins] with these replacements. Instead, most changes modulate the activity of most photoreceptors [and all opsins, rod and both cones]. A subset of the stimuli should achieve the same total quantal catch, but with different combinations of the two “primaries” of our LEDs. But we did not know *a priori* which stimuli these would be, instead determining it empirically (Figure 3).

Also, it would be conventional to determine for the stimuli the expected photoreceptor activation (the percentage photoreceptor contrast). The authors estimate the pigment density at the retinal locations of their stimuli, so presumably it should be possible to also report the cone (and rod) contrast for the stimuli in the various figures.

Because of the mixing of both opsins in many cones, it would be difficult to compute cone contrast per se, but it may be possible to compute “opsin” contrast as you suggest. We chose not do this because we could not be confident in the absolute calibration of our eye tracking system (see more extensive discussion of this below in the response, and also now in the manuscript in subsection “Analysis”). As such, we use eye tracking for the important controls of relative (not absolute) gaze elevation, and limit the use the absolute mapping between retinotopic to an estimation of relative opsin weight useful for putting our result in Figure 3 in context. Some error in the visuotopic gaze calculation would not effect that interpretation – that there is more rod contribution than expected. However, we were not comfortable stating stimulus intensities in terms of opsin contrast because of the potential inaccuracies of the method.

Figure 3 contains the main result, but it is a little confusing. I realize Figure 3 is a schematic, but it would help if there were some units, or at minimum, the location of (0,0). Is it the case that the (0,0) is at the center of the purple ellipse, and not at the bottom left corner (as would be typical for Cartesian coordinates)?

We have updated both Figure 3 and text describing it to clarify the result (subsection “Determination of relative short and middle wavelength band contributions at several retinotopic locations and comparison with predicted cone weights” and subsection “Can mice discriminate color changes?”). The schematic panels (A and D) of Figure 3 now include both units and an indication of the (0,0) points. Note that these axes are the change in each wavelength band, not the absolute value of each. So, at the (0,0) point the absolute hue, saturation, and lightness can be different, but there is no change (at the change time in the task) from whatever those absolute values are. At other points in these plots, the start and end colors can be different, but the magnitude and direction of the changes are the same. We also believe our stimulus diagrams for Figure 3 (and Figure 2) may have confused this issue. They are meant to indicate the relative change in each condition, not the absolute change, and this was not clear. We have therefore explicitly labeled the diagrams to not that they indicate that they are the relative change in each primary in each condition.

The text states that the major axis of the ellipse corresponds to a line of "constant saturation and lightness but varying hue" in HSV space. I don't think this is accurate. If the purple ellipse passes through (0,0), then it would pass through a point of zero saturation (at least if I am correctly interpreting the top panel of Figure 3).

We greatly appreciate the reviewers pointing out of this error. This is in fact a total misrepresentation of HSL color space and the data, and it has been removed. We believe the major axis of the ellipse corresponds to an axis of saturation changes, collapsed across hues.

Figure 3. It is not clear how "attempted compensation for relative short and middle band weights" was performed, and specifically how this is reflected in the graph (how is Figure 3 different from Figure 3?).

We have clarified the difference between Figure 3 and Figure 3 in both the Figure (by adding to the schematic in Figure 3) and text (subsection “Can mice discriminate color changes?”). The “attempted compensation” was made based on the observed short:middle weights at each elevation in Figure 3. We use the slopes of these lines to adjust the intensity of the green LED as a function of space. If the relative short:middle weights had remained the same after this adjustment, performance should have moved to the unity. This adjustment appears, not unexpectedly in retrospect, to have changed the relative rod:cone contribution and therefore the short:middle weight, though it did improve.

In Figure 3, is 50% correct equivalent to "chance"? I'm inclined to believe this is so, but then what is the interpretation for the significant performance below chance at several elevations (and for equiluminant stimuli of low color contrast)?

In our task the level of chance performance is not fixed by the task design, but can vary depending on strategy the mouse uses. It is fully possible for the mouse to regularly lick regardless of whether they have actually detected a change. In this regime, they would often restart the trial and enter the ‘time-out’, but every trial in which they (by chance) licked after a change (regardless of perceptibility) would result in reward. In this sense, the animal could reach 100% performance simply by ignoring the task and guessing, and thus “chance” could be 100%. By including catch trials, in which there was no actual change, we can estimate false positive rate at each elevation. There is therefore no “below chance” performance at any elevation (except a little dip below the false alarm rate at low color contrast for elevations=30º and 50º, the dark blue and pink lines in Figure 3, but these are small. One speculative interpretation of these dips is that below-threshold stimuli make the animal less likely to guess than no stimulus at all, possibly by adding uncertainty about whether there was a change or not, inhibiting the execution of a pure guess-after-some-time strategy).

And in the case where my interpretation is wrong, then how does one square the significant performance at the lowest elevation, at all color-contrast changes, with the performance on catch trials?

The observation that the false positive rate is very different at different elevations does require squaring. We believe that the increased false positive rate at the lowest elevation is due to a shift in guessing strategy by the mice correlated with the proportion of trials they can actually detect. At -10º, where the animal can’t detect many of the changes, the rate of guessing increases – the animal is willing to trade an increase in the number of time out penalties for an increase in false positive rewards. At the higher elevations, the animal is less willing to incur time outs, because a higher proportion of changes are detectable at these elevations. We went back and found the total number of rewards did not vary with elevation, even though the sensitivity varied, consistent with a change in strategy to maximize total volume of reward. We have added a brief discussion of this in the text (subsection “Can mice discriminate color changes?”).

Figure 3, shows that there is a set of stimuli defined by a ratio of S increment/M decrement (or M increment/S decrement) that is relatively invisible to the animals. The balance point of S/M is slightly different across the retina, with the more ventral visual field (dorsal retina) showing greater sensitivity to M than S, compared to the dorsal visual field. This suggests that the dorsal retina has a higher M:S ratio, which I believe is consistent with the cone-opsin distribution. Is this interpretation of the results correct? If so, aren't the behavioral data more consistent with the opsin distribution than the conclusions drawn by the authors?

This interpretation is correct, and we did not mean to suggest that our results are wholly inconsistent with the cone opsin distribution. Instead, we meant to highlight that the cone opsin distribution *alone* does not account completely for our results, both the relative increments/decrements required at each elevation (slopes of the ellipse major axes) and which elevations had the highest color sensitivity (change detectability along the major axes). We have changed some language throughout, and made this more explicit in the Introduction and in the Discussion section.

Reviewer 3 –I found the last section of the Results, with the description of Figure 3 and how hue sensitivity was measured, quite unclear in several parts (see my specific "minor" comments below). This description should be more detailed and much clearer, in order for a reader to fully grasp the meaning and implications of their findings.

We apologize for the lack of detail and clarity, here and elsewhere. We have added more detail and rewritten much of this section to improve clarity subsection “Determination of relative short and middle wavelength band contributions at several retinotopic locations and comparison with predicted cone weights” and subsection “Can mice discriminate color changes?”).

I have trouble understanding the experiments and data analysis to measure the sensitivity to hue changes Figure 3). Here is what I think I have understood and what I have not:The authors want to test if mice can detect changes along the diagonal line of perceived equiluminance (i.e., simultaneous and opposite changes of luminance in the two channels that are not detected by the rats as an overall change of luminance). Since, as they show in Figure 3, the perceived equiluminant line has not slope = 1, then the authors use this perceived line to measure hue sensitivity along it.… is this correct?

This is correct. We apologize for the lack of clarity in the Figure and our description of it. We have provided more detail and tried to clarify in the text and the figure.

Is this what they meant with Figure 3? Or, instead, did they actually show combinations of the two channels that are along the unity line? In any event, I don't understand why the shape of the purple bar changes when going from Figure 3 to Figure 3. Shouldn't the purple region have the same spear-like (not rectangular) shape of the region in Figure 3 (and of the data shown below in Figure 3)? More importantly, about the color matrixes shown in Figure 3: shouldn't these data just be the same of those already shown in Figure 3 (just taking those points along the equiluminant line)? What is the difference? Are these actually data resulting from different experiments? And if so, why?

The data in Figure 3 are not the same as that shown in Figure 3. The data in Figure 3 do indicate color sensitivity (along the equiluminance lines of not slope=1) at some elevations, but not at the lowest. However, we wanted to be sure that this observed color sensitivity was not due to some within-stimulus luminance gradient due the uniformity of the stimulus and gradient of opsins, even within the stimulus region, which was relativelity large (15º). So we took the equiluminance lines, obtained by fitting ellipses to the matrices in Figure 3, and adjusted the green stimulus uniformity. We then performed more testing sessions under these adjusted conditions, resulting in the data presented in Figure 3, with equiluminance lines closer to slope=1 and therefore less chance of any luminance gradient within the stimulus.

We have clarified the difference between panels Figure 3 in both the Figure (by adding to the schematic in Figure 3) and text (subsection “Can mice discriminate color changes?”). The “attempted compensation” was made based on the observed short:middle weights at each elevation in Figure 3. We use the slopes of these lines to adjust the intensity of the green LED as a function of space. If the relative short:middle weights had remained the same after this adjustment, performance should have moved to the unity. This adjustment appears, not unexpectedly in retrospect, to have changed the relative rod:cone contribution and therefore the short:middle weight, though it did improve.

Finally, are the curves shown in Figure 3 derived from the data matrixes of Figure 3? Are the curves an average of the data around the unity line? Or around the perceptual equiluminant line? If so, how many data points of the matrixes in Figure 3 are used to obtain the curves of Figure 3. In summary, this whole analysis needs far more details and explanations to be fully understandable?

The curves in Figure 3 are obtained along the lines in Figure 3. These lines in 3F which are from ellipse fit like those in Figure 3. There are 20 points in these curves.

2) Comments about measurement of eye position:The data suggest that the paradigm provides an accurate assessment of mouse visual ability. It is conceivable that the false-positive (false-alarm) response rate is not fixed, but changes depending on the detectability of the stimulus. It's not clear how this (or shifts in criterion) would impact the results.

The observation that the false-positive rate is not fixed, and that in our data it changes based on the overall detectability at that elevation, is astute. Indeed, we believe that the false-positive rate did rise for sets of stimuli with lower overall detectability – for example, the color-focused changes at -10º in Figure 3 (compare the green to other lines). Because we include catch trials as well as several other conditions that are likely below the detectability of the mouse, we believe the empirical measurement of false-positive rate is robust.

As you note, shifts in false-positive rate may impact estimation of the threshold. By using thresholds defined by a fit to the data, and allowing the offset and saturation parameters (subsection “Analysis”) the fit functions to be free should minimize the effect of the shifting false-positive rate. It is important that the false positive rate not be so high as to compress the range over which the true positive rates can be measured, but even in the case where performance was not distinguishable from the false positive rate (color discrimination at -10º, Figure 3), false-positive rate remained below 50%. As such, we do not believe the shifting false positive rate affects our conclusions.

The authors show that the mice do not have systematically different eye positions across trial types, but it is important to show that the experimental technique could recover differences in eye positions if these differences existed. What is the resolution of the eye tracking compared to the measured eye movements made by the animals in the experiment?

We believe the eye tracking system had more than enough resolution to recover potential differences in gaze elevation across stimulus conditions. Individual eye movements ranged from ~2 – 30º (see example movements added to Figure 1—figure supplement 1, and see below for more on the pixels-to-degrees conversion), and the eye tracker followed individual movements well large movements well. We have added an example period of eye tracking, which shows both the amplitude of these movements and the noise in the eye tracking. The frame-to-frame noise was ~1º [based on 95% of all frame-to-frame differences below 1º, for the right eye. Due to lower contrast in the eyetracking movie and higher magnification, the same measurement of the left eye was noiser, at < 4º. This has also been included in the eye tracking figure. As has been noted by others (Niell and Stryker, 2013), there were more, and larger, eye movements in the horizontal/azimuthal direction than vertical/elevation.

The authors mention that they have used an eye tracker, which, I believe, is essential to allow them to make accurate predictions of perceptual sensitivity, based on the known opsins' retinal distributions. However, as I far as I understand, the eye tracker was not used systematically, but only in some control sessions, whose data were not those used to obtain the sensitivity measures. I wonder then how the authors can be sure about the position of the mouse eye during the main experimental sessions (it may change form session to session or during a session, making their prediction unreliable).

We cannot be absolutely sure that the eye position was the same in the main behavioral testing sessions as it was in our eyetracking sessions, as you rightly note that we did not include eyetracking during these sessions. Nonetheless, we believe the eyetracking session accurately reflect the eye positions during other behavioral testing sessions. The eyetracking sessions were completed under conditions as close to the testing sessions as possible – using the same stimulus conditions in the same immersive visual environment at the same time of day. Given the consistency of the result with the cone opsin distribution and the overall lack of vertical eye movements, we don’t see reason to expect such a difference between the eye tracking sessions and the majority psychophysical testing sessions.

In addition, the authors do not provide any details (or any specific reference) about the eye tracking method they use, especially how they calibrated the eye tracker. Such a calibration is always challenging in rodents (since they do not saccade to target locations over the stimulus display). The authors report that they are able to obtain measurers in degrees, but they do not explain at all how they convert pixels to degree (these clarifications about the eye data are essential if we have to believe their predictions).

We apologize for the lack of detail and have added more to the Materials and methods section of the paper and several panels to the eye tracking figure. Calibration by guiding the mouse’s eye movements to specific known points on the display is indeed well beyond what we can do with our behavioral control of these animals. As such, the eye tracker is not explicitly calibrated. Instead we convert to degrees based on the geometry of the cameras, the position of the pupil relative to the corneal reflections of infrared LEDs, and an orthographic projection of the 2D eyetracking image onto a sphere. We now describe this conversion from pixels to degrees in the text (subsection “Analysis”), but it is also copied below:

To convert the tracked pupil positions from pixel coordinates in the eye tracking images to visuotopic coordinates, we first converted the pixel coordinates to spherical coordinates using an orthographic projection of the image onto a sphere with a radius of 1.16mm, the reported radius if the mouse eye. Because the mouse was positioned in the center of the stimulus enclosure (see the diagrams in Figure 1—figure supplement 1), which is also a sphere, the position of the pupil in these “eye” spherical coordinates can be converted to visuotopic coordinates by projecting the center of the “eye” sphere on to the visuotopic sphere. The reflection of an infrared LED off of the cornea indicated the center of each eye, and we use the placement of each mirror relative to the mouse (right eye: 51.2º in azimuth, 0º in elevation; left eye: 60º in azimuth and -10º in azimuth) to determine the direction of gaze. As such, the corneal reflection in each eye tracking movie indicates a known reference point in visuotopic space, determined by the relationship of the imaging plan to the eye sphere, and the difference between the corneal reflection and the pupil allows for estimation of gaze position in visual coordinates. Coordinates for eye position were extracted independently for each frame of the eye position movie. This calculation assumes that the center of both eyes are at the center of the visual stimulation dome, that each eye is a sphere, and that the movements of the eye are rotations about the center, none of which is strictly true. While left and right eye movies had different contrast and noise levels (see Figure 1—figure supplement 1), estimations of eye position in visual degrees brought disparate pixel measurement from each eye into good agreement with each other (compare y pixel measures to elevation in Figure 1—figure supplement 1). We believe this calculation to be accurate enough for comparison of relative eye positions and, given the small displacement of the eyes from the center (<1cm) of the dome (30 cm radius), reasonable for using published retinotopic distributions to estimate relative opsin distributions in visuotopic coordinates.

Because we cannot calibrate the system, we limit the eyetracking information to two uses: (1) a relative measurement of gaze, a crucial control that should not depend on the absolute accuracy of our calculation or the lack of calibration and (2) an interpretation of the opsin weight calculation that should be relatively insensitive to some error in this calculation. Even if our estimation of the retinotpoic-visuotpoic mapping was off in either direction, there would still be need to be more rod contribution to account for the observe middle-band sensitivity.

Finally, a specific reference for the eye tracking algorithm, to a technical white paper published on the Allen Institute website is provided in the text (the Allen Brain Observatory Visual Coding Overview v.4, June 2017 <http://help.brain-map.org/display/observatory/Documentation>).